# Strong Metal Support Effect of Pt/g-C_3_N_4_ Photocatalysts for Boosting Photothermal Synergistic Degradation of Benzene

**DOI:** 10.3390/ijms24076872

**Published:** 2023-04-06

**Authors:** Zhongcheng Huang, Xiaorong Cai, Shaohong Zang, Yixin Li, Dandan Zheng, Fuying Li

**Affiliations:** 1College of Environment and Safety Engineering, Fuzhou University, Fuzhou 350108, China; 2Institute of Innovation and Application, National Engineering Research Center for Marine Aquaculture, Zhejiang Ocean University, Zhoushan 316022, China; 3Donghai Laboratory, Zhoushan 316021, China; 4State Key Laboratory of Photocatalysis on Energy and Environment, Fuzhou University, Fuzhou 350116, China

**Keywords:** photodegradation, photothermal catalysis, carbon nitride, metal support effect

## Abstract

Catalysis is the most efficient and economical method for treating volatile organic pollutants (VOCs). Among the many materials that are used in engineering, platinized carbon nitride (Pt/g-C_3_N_4_) is an efficient and multifunctional catalyst which has strong light absorption and mass transfer capabilities, which enable it to be used in photocatalysis, thermal catalysis and photothermal synergistic catalysis for the degradation of benzene. In this work, Pt/g-C_3_N_4_ was prepared by four precursors for the photothermal synergistic catalytic degradation of benzene, which show different activities, and many tests were carried out to explore the possible reasons for the discrepancy. Among them, the Pt/g-C_3_N_4_ prepared from dicyanamide showed the highest activity and could convert benzene (300 ppm, 20 mL·min^−1^) completely at 162 °C under solar light and 173 °C under visible light. The reaction temperature was reduced by nearly half compared to the traditional thermal catalytic degradation of benzene at about 300 °C.

## 1. Introduction

Volatile organic compounds (VOCs), such as benzene, which is regarded as one of the most toxic and stable materials, are widely used in industrial production, building materials and automobile exhaust [1,2,3]. VOCs can cause irreversible effects on the environment and biology, such as teratogenicity, carcinogenicity and mutagenicity [4,5]. Therefore, it is necessary for current research to handle benzene with traditional methods, such as chemical absorption and combustion, ozonation, microbiological method and catalytic oxidation. In recent years, catalytic oxidation has become the main form to convert benzene to CO_2_ and H_2_O because of its great potential for the efficient degradation of benzene [6,7,8].

Catalytic oxidation has three types, including thermal catalytic oxidation, photocatalytic oxidation and photothermal catalytic oxidation [9,10,11]. Thermal catalysis oxidation can handle a large number of volatile organic compounds with high selectivity benefiting from thermal energy and noble metals, but it needs high-energy consumption. Photocatalytic oxidation can be carried out under mild conditions without secondary pollution; however, low efficiency and poisoning photocatalyst limit its application [12,13]. It is of great difficulty to convert the high-concentration VOCs under mild conditions by photocatalysis or thermal catalysis alone. Photothermal catalytic oxidation (PTO) overcomes the shortcomings of high-energy consumption and low efficiency of traditional catalytic reactions [14,15,16]. The introduction of heat energy can promote carrier migration, accelerate mass transfer and facilitate the dissociation of reactants. On the other hand, in thermodynamics, regulating the reaction temperature is beneficial to regulate the redox potential of the reaction and to expand the application range of narrow gap semiconductors [17,18,19]. Therefore, PTO has attracted a wide concern from researchers in benzene degradation in recent years. Photocatalysts with thermal effects can completely degrade VOCs under the action of photothermal coupling, which offers a new way for the treatment of high-concentration VOCs [10,19,20,21].

Currently, the most widely used photocatalysts in photothermal catalytic degradation of VOCs are TiO_2_-based composite materials and rare earth metal salts. For example, Fang et al. used BiVO_4_/TiO_2_ for photothermal synergistic catalysis, which makes benzene completely converted at low temperatures [22]. Yang et al. prepared Pt/SrTiO_3_ for the photothermal synergistic catalysis of toluene and the most excellent treatment effect was obtained at 150 °C [23]. However, the application of TiO_2_ is limited by weak visible light absorption. In 2009, Wang et al. successfully synthesized and applied carbon nitride (g-C_3_N_4_) with a visible light response in water splitting [24]. Subsequently, g-C_3_N_4_ has been applied widely for its excellent reducibility, chemical stability and lack of toxicity. Common g-C_3_N_4_-based materials, such as WO_3_/g-C_3_N_4,_ CoS_2_/g-C_3_N_4_, TiO_2_/g-C_3_N_4_ [25,26,27,28,29], etc., have made outstanding progress in the application of photocatalysis. However, the application of low-temperature PTO is rarely reported.

Profiting from its stable planar electron π-conjugated structure and large specific surface, g-C_3_N_4_ can conduct electrons and adsorb VOCs efficiently. Although it has good reducibility, the development of g-C_3_N_4_ in the gas–solid phase catalytic oxidation of benzene is sluggish because of the low conversion rate and mineralization rate. Limitations include the weak oxidizing property of g-C_3_N_4_ and the high stability of saturated C–H bonds in benzene [30,31,32]. Many strategies have been adopted to improve the efficiency of g-C_3_N_4_ in benzene degradation, such as doping, cocatalysts loading and composites construction [33,34,35,36,37,38]. For example, Mamari et al. synthesized ZnO/g-C_3_N_4_ composites which could degrade benzene by visible light [37]. Zou et al. prepared BiPO_4_/g-C_3_N_4_ through a hydrothermal method, including calcination; the improved photoactivity could be attributed to the effective separation of photogenerated charge carriers for a 73% degradation rate of benzene [7]. However, g-C_3_N_4_-based materials are rarely used in photothermal synergistic catalysis.

Research has revealed that the loading of noble metals cocatalysts, such as Pt, Au and Ag, etc., could enhance the photocatalytic performance. On one hand, the appropriate work function enables noble metal cocatalysts to build efficient charge separation between semiconductor and cocatalysts through the Schottky junction or Ohmic contact construction within the metal-semiconductor structure. However, on the other hand, they were in favor of improving visible light absorption influence and promoting photoinduced charge separation and active oxygen species production (·O_2_^−^) using the surface plasmon resonance effect (SPR) [39,40,41,42]. The strong interaction of metal–support is mainly through the formation of negatively charged noble metal nanoparticles (NPs) on the carrier and the influence of chemically adsorbed oxygen. Driven by light (heat), electrons will continuously flow from catalysts to the noble metal until their Fermi energy levels are equal. In the space charge layer, the metal surface will obtain excess negative charges, while the catalyst’s surface will appear as excess positive charges to catalyze the degradation of VOCs. Meanwhile, the addition of heat can promote the SMSI effect to promote the oxygen activation in the thermaocatalytic oxidation process, so that the pollutants can be degradated quickly on the surface of the carrier and the precious metals [43,44,45,46].

In this work, g-C_3_N_4_ was prepared by four different precursors through thermal polymerization and by loading Pt NPs uniformly using photodeposition. The characteristic of Pt/g-C_3_N_4_ was discussed in the terms of morphologic structure, characterization and photothermal synergistic catalysis activity via various techniques. When Pt NPs were supported on the surface of g-C_3_N_4_, there was also a strong metal–support interaction effect, which could enhance the photothermal catalytic degradation of benzene. The plasma effect of Pt can promote the adsorption of O_2_ and converts it into superoxide radicals to participate in the ring-opening degradation of benzene, making the performance of g-C_3_N_4_ gas–solid phase photothermal synergistic catalysis better than traditional methods. The performance of the prepared CNs was evaluated by the degradation of benzene using photothermal synergistic catalysis. Among them, the Pt/g-C_3_N_4_ prepared from dicyanamide (named PDA) showed the highest activity. It could convert benzene (300 ppm, 20 mL·min^−1^) completely at 162 °C (irradiated by solar light) and 173 ℃ (irradiated by visible light). Whereas the reaction temperature of the traditional thermal catalytic degradation of benzene happened at about 300 °C. Meanwhile, PDA showed excellent stability after a four-cycle reaction. The mechanism was also explored using DRS, electrochemical experiments and EPR.

## 2. Results and Discussion

The crystal structure of the sample was characterized using XRD. In Figure 1a, the XRD diffraction peaks demonstrate that Pt/g-C_3_N_4_ samples synthesized from different precursors had different intensity peaks at 13° and 27.5°, corresponding to (100) and (002) planes of g-C_3_N_4_, respectively [47,48]. No other impurity peaks appeared indicating that the synthesized samples were pure without other impurities. Due to the structural differences of precursors, there are obvious differences in morphology, interplanar packing space and crystallinity of g-C_3_N_4_ synthesized in the thermal polymerization process. Besides, the possible incompletion of condensation generated during the polymerization of different precursors could be considerably affected by the XRD characteristic peaks. Taking g-C_3_N_4_ prepared with urea as an example, when the temperature increased, the functional group of ammonia detached and formed isocyanic acid which could condense with another urea molecule to form a biuret. Then, the biuret forms cyanuric acid by a nucleophilic addition reaction, which in turn is converted to melamine. Melamine is polymerized by deamination to form melon, which can then form the triazine ring structure g-C_3_N_4_ by thermal polymerization. The degrees of stacking of g-C_3_N_4_ are different from precursors under thermal polymerization, leading to differences in their crystallinity.

In the XRD pattern, the half-peak width reflects the crystallinity of the crystal, which affects the photothermal synergistic catalysis. The sharper the peak is, the higher the crystallinity of the sample, which is more in favor of photogenerated electron transfer. The half-peak width of PMA and PDA was narrower than that of PCA and PUR. The FT-IR spectra of Pt/g-C_3_N_4_ are shown in Figure 1b. The peaks at 1200–1600 cm^−1^ belonged to the vibration of the CN ring absorption, and the peaks at 805 cm^−1^ were mainly attributed to the breathing vibration of the triazine. The absorption peaks located at 3000–3300 cm^−1^ are mainly attributed to O–H, N–H vibration, which indicated the presence of uncondensed amino functional in the CN. Besides, the peaks at 1284 and 1213 cm^−1^ were associated with the vibration modes of N–(C)_3_ and C–N–C in the heptazine rings [36,47]. In addition, the characteristic peaks of PCA could not be found at 1213 and 1442 cm^−1^ and the peaks intensity of PCA were weaker than that of other CNs. This indicates that there are more defects in the structure of heptazine rings of PCA, which leads to the reduction in crystallinity in PCA. The information obtained from FT-IR spectra is consistent with XRD results as well.

The SEM images shown in Figure 2a were used to observe the morphologies of the Pt/g-C_3_N_4_. The prepared product was predominantly composed of crystals with several microns. The g-C_3_N_4_ prepared from dicyandiamide, cyanamide, and melamine were all micro-sized block structures shown in Figure 2a, while the g-C_3_N_4_ obtained using urea, was composed of numerous stacked nanosheets (Figure 2b). However, there were no Pt nanoparticles that could be observed maybe because of the small size of Pt. For an in-depth understanding of the crystal structure and composition, the Pt/g-C_3_N_4_ synthesized with dicyandiamide (PDA) was observed using TEM. In Figure 2c, the sample was a block with a size of 2 μm. As shown in Figure 2d, Pt NPs were distributed uniformly on the surface of PDA with a particle size of about 2 nm. The interplanar spacing of Pt NPs was 2.26 Å, corresponding to the (111) plane of Pt, where was the insert image with red frame shown in Figure 2e [21,31,49]. The EDX elemental mapping (Figure 2e,f) showed that C, N and Pt elements were all detected and displayed a uniformity of Pt dispersion in PDA.

XPS analysis was applied to elucidate the chemical composition and oxidation state of PDA. As shown in Figure 3a, the sample was mainly composed of C, N, O and Pt elements. The percentage of C, N and H in PDA was 61.67%, 34.83% and 2.10%, respectively. The two formers were the main component of g-C_3_N_4_, while O was a part of oxygen introduced in contact with the air during thermal polymerization. The XPS spectrum of Pt 4f in Figure 3b was divided into four peaks, the peaks located at 72.8 eV and 76.0 eV contributed to Pt 4f_7/2_ and the other one at 74.8 eV and 78.1 eV corresponding to Pt 4f_5/2_ spin–orbit. In Figure 3c, the three XPS peaks of C 1s were located at 288.2, 286.4 and 284.8 eV, which was attributed to the N–C=N of the triazine ring, C–O=C bond and C=C group of g-C_3_N_4_, respectively [50]. The characteristic peaks of N 1s in Figure 3d located at 398.3 and 400.0 eV corresponded to C–N=C and N–(C)_3_ groups, respectively. Both of them together with the sp^2^–C (N–C=N) make up the C_6_N_7_ units [51]. The last one of N 1s at 401.1 eV arose from the uncondensed C–N–H groups of g-C_3_N_4_. The XRD, FT-IR and XPS analysis of PDA indicated that the main structure of g-C_3_N_4_ was still well preserved after loading Pt NPs.

The specific surface area is an important factor affecting the catalytic performance. The BET surface area is shown in Figure 3e. The adsorption and desorption curves of PDA, PCA, PMA and PUR were the typical type IV isotherm of the H_3_ hysteresis loop. Although PUR has the largest specific surface area, the low density and poor heat transfer properties make it difficult to achieve a good result in the gas–solid phase photothermal synergistic catalysis process at a higher airspeed. Under the same conditions, PDA with a larger specific surface area than that of PMA can promote contact between the catalysts and pollutants. Since the tests were carried out in the presence of air or an oxygen–benzene mixture, thermogravimetric analysis was carried out. According to the thermogravimetric analysis (Figure 3f), it can be seen that the decomposition temperature varies greatly under N_2_ and O_2_ conditions. In the presence of oxygen, the decomposition of PDA occurred at about 480 °C. However, this work was carried out in relatively mild conditions, far below the decomposition temperature of PDA.

The performance of these Pt/g-C_3_N_4_ was measured using photothermal synergistic catalysis of benzene under 150 °C and solar light conditions. The experimental results are shown in Figure 4a,b, It can be seen that there are certain differences in the conversion/mineralization rate; PDA shows excellent performance, which could reach more than 95% at 150 °C and converse completely at 162 °C. In addition, each catalyst showed the same trend in the process of photothermal synergistic catalysis of benzene, that was, the conversion/mineralization rate gradually increased and then stabilized with the extension of time. According to the comprehensive analysis of conversion rate and mineralization rate, PDA was the best one in the PTO of benzene (Figure 4a,b).

To explore the influence of the photo/thermal effect on the performance of PDA, the photothermal-coupled degradation of benzene under only light, only heating, photothermal synergistic catalysis with solar light and visible light were also investigated (Figure 4c,d). PDA cannot degrade benzene only under light conditions, and the conversion efficiency was only 5% at 30 °C. Although PDA can generate photogenerated carriers under illumination, due to the excessively high-space velocity, benzene cannot be rapidly and completely oxidized on the catalyst surface. In the case of only heating, the C% increased significantly with the increment of temperature and benzene was converted completely at 190 °C. When light and heat existed at the same time, benzene could be converted completely at 162 °C under solar light, and at 173 °C under visible light (Figure 4e). The combination of light and thermal catalysis significantly reduced the conversion temperature and improved the conversion efficiency of PDA. The four-round cycle experiments with 8 h of photothermal synergistic catalytic degradation of benzene were carried out at 162 °C in Figure 4f. It can be seen that PDA still maintains a high conversion (mineralization) rate after the loop experiment, which indicated that the performance of the photothermal stability of the catalyst was strong.

DRS was used to measure the light absorption of the catalyst. As shown in Figure 5a, the light absorption band edges of carbon nitride with platinum nanoparticles showed an obvious red shift compared to pure g-C_3_N_4_. The red shift broadens the visible light absorption range of CN. The DRS curve of PUR and PCA had an upward tail, indicating that a mass of defects existed on the surface of PUR and PCA, which was harmful to benzene degradation. The separation and recombination rates of photogenerated electron–hole pairs were also investigated using photoluminescence emission spectra (PL). As depicted in Figure 5b, the PL intensity of PDA was the lowest, showing that PDA had the lowest charge recombination rate.

The transient photocurrent test was used to characterize the charge transport and separation efficiency of the sample and used the three-electrode method to test the photocurrent response value of the PDA sample. It is found that the PDA sample has the highest under solar light irradiation, which indicates that PDA can generate more e^−^/h^+^ pairs and that the photogenerated carrier flux is fast, which corresponds to the activity of catalyzing the degradation of benzene. Simultaneously, their impedances were analyzed. The smaller the radius of the Nyquist arch in the electrochemical impedance spectrum, the lower the migration resistance of the photogenerated carriers. In Figure 5c, the order of radii was PCA > PUR > PMA > PDA, which indicated that PDA had the highest charge transfer efficiency. The photocurrent experiment in Figure 5d also showed the same conclusion, which meant that PDA could generate more carriers under the excitation of light. The enhanced photocurrent was due to the SPR effects of Pt NPs in PDA, which further promoted photoinduced charge separation. This was the reason why PDA made the greatest performance in the photothermal degradation of benzene.

The reactive oxidation species have a decisive effect on the PTO of benzene. They were detected using EPR and the data are shown in Figure 5e,f. In dark conditions, the signals of radicals were very weak. After 300 s of light irradiation, the characteristic peaks of DMPO trapped ·OH and ·O_2_^−^ adducts appeared in the aqueous suspension of PDA. EPR data indicated that the main oxidative species in the benzene degradation experiment using PDA photocatalyst were ·OH and ·O_2_^−^. Based on the above analysis, a possible mechanism was put forward in Figure 1. In the photothermal synergistic catalysis process, photoexcitation can make the matrix carbon nitride generate excited electrons so that the electrons continuously flow to the Pt NPs. Meanwhile, thermal catalysis not only reduced the activation energy required for the reaction but also generated the hot electrons. It is rapidly transferred to the active site to participate in the reaction so that the O_2_ adsorbed by Pt in the reaction could be quickly converted into ·O_2_^−^, the OH^-^ was converted to ·OH and both of them promoted the degradation of benzene. Combined with the activity data of photothermal catalytic degradation of benzene and the tested results, a conclusion can be drawn. The SMSI effect of Pt/g-C_3_N_4_ led to the formation of chemisorbed oxygen and negatively charged Pt NPs, which promoted oxygen activation in the thermocatalytic oxidation process and SPR effects of Pt NPs in the photocatalytic oxidation process [52,53,54]. The SPR effects further promoted photoinduced charges separation and the ·O_2_^−^ formation [39,42]. Thus, the efficiency of photothermal synthesis catalytic degradation of benzene improved.

## 3. Materials and Methods

Materials: All chemicals were used without further purification.

Analytical-grade Chloroplatinic Acid (H_2_PtCl_6_), cyanamide (CA), dicyanamide (DA), urea (UR) and melamine (MA) were purchased from Sinopharm.

### 3.1. Preparation of g-C_3_N_4_ and Pt/g-C_3_N_4_

Typically, 10 g of the precursor was loaded into a crucible with a cover. Then, the crucible was put into a maffle and heated to 550 °C for 4 h at a rate of 5 °C/min. After cooling, the samples were ground to obtain powdered g-C_3_N_4_.

Subsequently, Pt NPs were loaded on the surface of g-C_3_N_4_ using photodeposition with H_2_PtCl_6_ which was named PCA, PDA, PUR and PMA according to different precursors. The obtained sample was dried and sieved to 50–70 mesh for use.

### 3.2. Characterization

The morphology and elemental mapping images of catalysts were characterized using field emission scanning electron microscopy (SEM, SU8000, Japan Hitachi Co., Tokyo, Japan) and a transmission electron microscope (TEM, Talos F200s, FEI Co., Hillsboro, OR, USA). X-ray diffraction (XRD, D8 Advance, Germany Bruker Co., Billerica, MA, USA) patterns using Cu Kα radiation were used to characterize the crystal structure of samples. The surface chemical state of the catalysts was characterized by X-ray photoelectron spectroscopy (XPS, ESCALAB 250Xi, Thermo Scientific Co., Waltham, MA, USA, the standard peak is the C 1s peak at 284.8 eV). The UV-vis diffuse reflectance spectra of the samples were measured by an integrating sphere UV-vis spectrophotometer (DRS, Cary 5000 Scan Spectrophotometer, Agilent Cross Lab., Palo Alto, CA, USA). Photoluminescence (PL) was performed on Fluorolog-3 (HORIBA Co., Tokyo, Japan) to research the recombination of charge carriers of catalysts. Electrochemical impedance spectroscopy (EIS) of the samples was tested using an electrochemical workstation (VSP–300, France Bio–Logic Co., Saône-et-Loire Department, France), which contained a three-electrode system. The working electrode was the FTO glass with catalysts (5 mg catalyst in 0.9 mL DMF and 0.1 mL Nafion with 0.25 cm^2^ active area), the reference electrode was an Ag/AgCl electrode and the counter electrode was Pt wire. The electrolyte was a 0.2 M Na_2_SO_4_ aqueous solution.

### 3.3. Photothermal Synergistic Catalytic Degradation of Benzene

The photothermal synergistic catalytic degradation of benzene takes place in a three-way reactor, which includes an inlet, an outlet and an external thermocouple. In addition, benzene was generated using a gas bubbling device, which makes benzene diluted with oxygen so that the inlet concentration of benzene was about 300 ppm, 20 mL·min^−1^. Benzene passed through a quartz reactor equipped with catalysts continuously. Before the reaction, benzene flowed continuously through the catalyst for 12 h in darkness to establish an adsorption-desorption equilibrium. Then, reacted under photothermal conditions for 8 h to investigate the activities of photothermal synergistic catalytic degradation of benzene.

The temperature of the reaction system was controlled using an external magnetic stirrer and a thermocouple. The simulated sunlight and visible light source were provided by an external 500 W Xenon lamp. The gas product type and contents were detected using an Agilent 7890A online gas chromatograph.

A total of 0.15. g catalyst was put into a quartz three-way reactor and then the catalyst was subjected to adsorption and desorption treatment under dark conditions until the concentrations at the outlet were stable. Then, the light source and heater were turned on to stabilize the temperature of the quartz reactor at a fixed value. The concentration of benzene and carbon dioxide were detected using online gas chromatography, and the conversion/mineralization rate was calculated to evaluate the performance of different catalysts. The conversion rate (C%) and mineralization rate (M%) are calculated as follows.
(1)$$C\%=\frac{\left(C_{0}-C\right)}{C_{0}}\times 100\%$$
(2)$$M\%=\frac{{\left[CO_{2}\right]}_{prod}}{\left[6\times \left(C_{0}-C\right)\right]}\times 100\%$$

*C*_0_ is the concentration of C_6_H_6_ when the absorption and desorption equilibrium is reached, *C* is the concentration of C_6_H_6_ remaining after degradation and [*CO*_2_]*_prod_* refers to the concentration of CO_2_ generated during the photothermal synergistic catalytic degradation process.

## 4. Conclusions

In conclusion, g-C_3_N_4_ was synthesized by a simple thermal polymerization process using four different precursors and by loading Pt NPs using photo-deposition. Among them, PDA had the highest activity in the degradation of benzene after loading Pt NPs. The mechanism of benzene degradation was also discussed. Light can promote PDA to generate photogenerated electron–hole pairs, then, electrons flow to Pt NPs continuously. Heat would reduce the activation energy of the catalyst to degrade benzene, and at the same time, accelerate the transfer of electrons and capture of electrons by O_2_, which will accelerate the progress of the reaction. In photothermal synergistic catalysis, the thermal action of photoexcited electrons accelerates the flow of electrons and promotes the degradation of benzene. Therefore, benzene could be degraded under milder conditions, and it also has high activity under visible light. This work opens up the use of g-C_3_N_4_ for the efficient degradation of benzene under visible-light and solar light photothermal conditions.

## Data Availability

Not applicable.

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
