# Peer review of "Strong Metal Support Effect of Pt/g-C3N4 Photocatalysts for Boosting Photothermal Synergistic Degradation of Benzene"

_ijms, 2023, doi:10.3390/ijms24076872_

Round 1
Reviewer 1 Report
In this article, Zhongcheng Huang et al. investigated the photothermal synergistic degradation of benzene based on strong metal supportive effect of Pt/g-C3N4 photocatalyst. Although the degradation of VOC compounds is not a new concept, the authors have given their best in exploring the possible ways to degrade the benzene and the total theme of experimental work is very nice. The stated results are interesting and the manuscript is also well organized/written, which will likely to attract attention and impact in VOC degradation and photocatalytic applications. As such, I believe that the manuscript is suitable for publication; in which following minor amendments need to be clarified before this work can be finally accepted in IJMS journal:
1. In page NO1, line NO:13 the word in engineer, not appropriate it could be engineering. Please recheck it.
2. What is full−light? White light or solar light?
3. Line No: 142 -and desorbed under dark conditions for 12 h, and then illuminated for 8h. Sentence needs to be modified.
4. Is the method of synthesis of Pt/g-C3N4 is doping or embedding or decorating? Since it is important in charge transfer. Any Schottky/ohmic behavior that authors have noticed.
5. Synergistic effect of Pt/g-C3N4 should be further explained detailedly as their possible optoelectrical properties in the introduction and mechanism sections.
6. From the XRD patterns, it seems there are slight shifts in the data. What could be the reason for it? What might be the reasons for the FWHM width expansion and decrease in peak intensity? Explain elaborately.
7. Please supplement the zoomed view of Pt d-spacing from Fig. 2e for better clarity.
8. Provide the elemental percentage in the photocatalyst.
9. How about long-term stability?
10. What are the main factor for enhancing the photocurrent and high degradation conversion effect for PDA. Explain in more detailly.
11. Authors have explained SPR effect of metal nanoparticle but in the mechanism section the role of Pt-SPR effect was not discussed please add this effect also for better understanding.
12. Some of the refences are not up to the mark and here are some of the latest refences which are latest and related your work and hence it could be read and cite in necessary places to further support your work. Latest references must be cited to claim the necessity of your research. 10.1021/acsami.8b03832, 0.1016/j.snb.2022.133140, 10.1002/cey2.293, https://www.sciopen.com/article/10.1007/s12274-023-5472-x.
Author Response
Reviewer 1
Comment 1: In page No.1, line No.13 the word in engineer, not appropriate it could be engineering. Please recheck it.
Response 1: Thanks for the comments. Line 13, ‘engineer’ was changed to ‘engineering’ (marked in yellow color).
Comment 2: What is full−light? White light or solar light?
Response 2: Thanks for your suggestion. The ‘full-light’ refers to ‘simulated solar light’. In case of misunderstanding, the ‘full-light’ in the article has been changed to ‘solar light’ (marked in yellow color in the revised manuscript).
Comment 3: Line No: 142 -and desorbed under dark conditions for 12 h, and then illuminated for 8h. Sentence needs to be modified.
Response 3: Thanks for your comments, which sentence have been modified in the manuscript “Before the reaction, the continuously benzene flowed through the catalyst for 12h in the darkness to establish adsorption-desorption equilibrium. And then reacted under photothermal conditions for 8h to investigate the activities of photothermal synergistic catalytic degradation of benzene.” (marked in yellow in page No.3, line 143-146).
Comment 4: Is the method of synthesis of Pt/g-C3N4 is doping or embedding or decorating? Since it is important in charge transfer. Any Schottky/ohmic behavior that authors have noticed.
Response 4: Thank you for your question. Pt nanoparticles were reduced to atoms and decorated onto the catalyst surface by in-situ photodeposition, which is inclined to a surface modification method. Recent studies showed that the loading of the precious metals, such as Pt, Au and Ag, could enhance the photocatalytic performance of the conventional photocatalysts through facilitating photoinduced charges separation by Schottky barrier and improving visible-light absorption by the surface plasmon resonance (SPR) effect (Ref. Chemosphere 2020, 249, 126096; Nanoscale Adv. 2022, 4(12), 2608-2631).
Comment 5: Synergistic effect of Pt/g-C3N4 should be further explained detailedly as their possible optoelectrical properties in the introduction and mechanism sections.
Response 5: Thanks for your suggestions. Strong metal-support interaction in g-C3N4 supported by Pt nanoparticles, led to the formation of chemisorbed oxygen and negatively charged Pt NPs, which promoted oxygen activation in thermocatalytic oxidation process (TOC) and surface plasmon resonance effects of Pt NPs in photocatalytic oxidation (POC), visible-light absorption, photoinduced charges separation, and active oxygen species production (·O2-). The revised sections have made in the manuscript (marked in yellow, line No.318-320)
Comment 6: From the XRD patterns, it seems there are slight shifts in the data. What could be the reason for it? What might be the reasons for the FWHM width expansion and decrease in peak intensity? Explain elaborately.
Response 6: Thanks for your suggestion. Due to the structural differences of different precursors, there are obvious differences in morphology, interplanar packing space and crystallinity of g-C3N4 synthesized in the thermal polymerization process. For example, when UREA is used as the precursor to prepare g-C3N4, carbon nitride tends to form spatial layered structure. However, when DCDA and MA used as the precursor, the g-C3N4 trends to form bulk phase structure, which may lead to a difference in crystallinity, resulting in a slight shift of XRD peaks. The modified place has been marked in yellow. (Line No.171-174)
Comment 7: Please supplement the zoomed view of Pt d-spacing from Fig. 2e for better clarity.
Response 7: Thanks for your comments. We have increased the zoomed view of Pt d-spacing in Fig. 2e. The modified image has been replaced in the manuscript.
Comment 8: Provide the elemental percentage in the photocatalyst.
Response 8: Thanks for your comments. The percentage of C, N, H in PDA were 61.67%, 34.83%, 2.10%, respectively. The revised sections have made in the manuscript (marked in yellow, line No.220)
Comment 9: How about long-term stability?
Fig. R1 Conversion/mineralization rate of four−cycle experiments of PDA under 162 ℃ and solar light condition.
Response 9: As shown in Fig. R1, the four-cycle experiments under photothermal condition at 162 ℃ were carried out to investigate the long-term stability. The results show that PDA maintains excellent stability in long-term photothermal catalysis. The relevant section was shown in Fig. 4f in the revised paper. (Line No. 265-269).
Comment 10: What are the main factor for enhancing the photocurrent and high degradation conversion effect for PDA. Explain in more detailly.
Response 10: Thanks for your suggestion. The enhance photocurrent was due to the formation of Pt/g-C3N4 Schottky junction which promote the separation of photogenerated carrier. Driven by heat and light, electrons will continuously flow from catalysts to Pt NPs, which may react with adsorbed oxygen to generate active oxygen species production and promote the reaction. The relevant section has been marked in yellow. (Line No.300-301, 316-320)
Comment 11: Authors have explained SPR effect of metal nanoparticle but in the mechanism section the role of Pt-SPR effect was not discussed please add this effect also for better understanding.
Response 11: Thanks for your comments. Combined with the activity data of photothermal catalytic degradation of benzene and the tested results, a conclusion can be drawn. The SMSI effect of Pt/g-C3N4 led to the formation of chemisorbed oxygen and negatively charged Pt NPs, which promoted oxygen activation in thermocatalytic oxidation process and SPR effects of Pt NPs in photocatalytic oxidation process. The SPR effects further promoted photoinduced charges separation and the ·O2- formation. Thus, the efficiency of photothermal synthesis catalytic degradation of benzene improved obviously. The revised sections have made in the manuscript (marked in yellow, line No.316-323)
- Some of the refences are not up to the mark and here are some of the latest refences which are latest and related your work and hence it could be read and cite in necessary places to further support your work. Latest references must be cited to claim the necessity of your research. 10.1021/acsami.8b03832, 10.1016/j.snb.2022.133140, 10.1002/cey2.293, https://www.sciopen.com/article/10.1007/s12274-023-5472-x.
Response 12: Thanks for the valuable suggestion. These papers have been cited as listed in Ref. 13, 31, 52, 53, 54 in the revised manuscript in References (marked in yellow color).
Ref:
[1] Sourabh S C, Jeffery A A, Sreya Roy C, et al. Antipoisoning catalysts for the selective oxygen reduction reaction at the interface between metal nanoparticles and the electrolyte[J]. Carbon Energy, 2023, e293.
[2] Cheng K, Zhu K, Liu S, et al. A Spatially Confined g-C3N4–Pt Electrocatalyst with Robust Stability[J]. ACS Applied Materials & Interfaces, 2018, 10(25): 21306-21312.
[3] Kedhareswara Sairam P, Sayandeep G, Nagabandi J, et al. Boron doped g-C3N4 quantum dots based highly sensitive surface acoustic wave NO2 sensor with faster gas kinetics under UV light illumination[J]. Sensors and Actuators B: Chemical, 2022, 378, 133140.
[4] Hongqiang X, Lan S, Yiwei Z, et al. Surpassing Pt hydrogen production from {200} facet-riched polyhedral Rh2P nanoparticles by one-step synthesis[J]. Applied Catalysis B: Environmental. 2023, 330, 122645.
List of changes:
- Line 13, ‘engineer’ reminds to ‘engineering’; Line 92, ‘metals’ was added; Line232, ‘factors’ reminds to ‘factor’; Line 233 ‘were’ revises to ‘was’. And some other mistakes were also corrected and marked.
- In case of misunderstanding, the ‘full-light’ in the article has revised to ‘solar light’.
- Line 82-83, “And on the other hand, they are in favor of improving visible light absorption influenced and promot photoinduced charges separation and active oxygen species production (·O2-) by the surface plasmon resonance effect (SPR)”.
- Line 90-91, Meanwhile, the addition of heat can promote the SMSI effect to promoted the oxygen activation in thermaocatalytic oxidation process, so that the pollutants can be degradated quickly on the surface of the carrier and precious metals.
- Line 143-146, Before the reaction, the continuously benzene flowed through the catalyst for 12h in the darkness to establish adsorption-desorption equilibrium. And then reacted under photothermal conditions for 8h to investigate the activities of photothermal synergistic catalytic degradation of benzene.
- Line 171-173, “Due to the structural differences of precursors, there are obvious differences in morphology, interplanar packing space and crystallinity of g-C3N4 synthesized in the thermal polymerization process.” was added in the revised manuscript.
- Line 199 and Line 279, the Fig. 2 and Fig.5 has revised in the manuscript.
- Line 300-303, “The inorganic-organic heterostructure lead the formation of negatively charged Pt NPs and promote the separation of photogenerated carrier which could react with adsorbed oxygen to generate active oxygen species production and promote the reaction.” was added in the revised manuscript.
- Line 314-321, “Combined with the activity data of photothermal catalytic degradation of benzene and the tested results, a conclusion can be drawn. The SMSI effect of Pt/g-C3N4 led to the formation of chemisorbed oxygen and negatively charged Pt NPs, which promoted oxygen activation in thermocatalytic oxidation process and SPR effects of Pt NPs in photocatalytic oxidation process. [53, 54] The SPR effects further promoted photoinduced charges separation and the O2- formation. [39,42] Thus, the efficiency of photothermal synthesis catalytic degradation of benzene improved obviously.” This sentence was added into the revised manuscript according to the reviewer’s comments.
- Line 322-325, “We can draw the conclusion that the SMSI effect in g-C3N4 supported by Pt NPs, led to the formation of chemisorbed oxygen and negatively charged Pt NPs, which promote oxygen activation in TOC process and SPR effects in POC process for photothermal synthesis catalytic degradation of benzene.” This sentence was added into the revised manuscript according to the reviewer’s comments.
- Reference 13, 31, 52, 53,54 were added in the revised manuscript.
Reviewer 2 Report
Here are my comments:
1-The English of the manuscript should be improved a lot.
2- Consider shortening the title.
3- Several vague terms can be found in the text, for example, " full−light" which should be revised by the authors.
4- The presented data does not have error bars. The statistical analysis should be done on the data graphs presented.
5-Be consistent in using abbreviations ... e.g., in text authors used "Fig. x" and in captions "Figure x".
6- Few old references need to be replaced with new studies ... for examples 10.1016/B978-0-12-818806-4.00010-3
Author Response
Comment 1: The English of the manuscript should be improved a lot.
Response 1: Thank you for your suggestion, we have made corrections on the experimental section and the manuscript was also went thought by native speaker. The modified place has been marked in yellow.
Comment 2: Consider shortening the title.
Response 2: Thanks for your suggestions. The title has been revised to “Strong metal support effect of Pt/g−C3N4 Photocatalysts for Boosting photothermal synergistic degradation of benzene”.
Comment 3: Several vague terms can be found in the text, for example, " full−light" which should be revised by the authors.
Response 3: Thanks for your comments. The ‘full-light’ in the article refers to ‘simulated solar light’. In case of misunderstanding, the ‘full-light’ in the article has been changed to ‘solar light’ (marked in yellow color in the revised manuscript).
Comment 4: The presented data does not have error bars. The statistical analysis should be done on the data graphs presented.
Fig. R1 (a) conversion rate, (b) mineralization rate of benzene for PDA under only heat, visible−light photothermal or solar light photothermal conditions, (c) conversion/mineralization rate of four−cycle experiments of PDA under 162 ℃ and solar light condition.
Response 3: Thank you for your comments. As shown in Fig. R1, the error bar was added. (marked in yellow color in Fig. 4c, d and f in the revised manuscript)
Comment 5: Be consistent in using abbreviations ... e.g., in text authors used "Fig. x" and in captions "Figure x".
Response 5: Thank you for your comments, we have corrected the relevant part in the manuscript.
Comment 6: Few old references need to be replaced with new studies ... for examples 10.1016/B978-0-12-818806-4.00010-3.
Response 6: Thanks for the valuable suggestion. These papers have been cited as listed in Ref. 13, 31, 52, 54 in the revised manuscript in References (marked in yellow color).
Ref:
[16] Taghizadeh A, Taghizadeh M, Sabzehmeidani M M, et al. Electronic structure: From basic principles to photocatalysis. Interface Science and Technology. 2021, 32, 1-53.
[52] Sourabh S C, Jeffery A A, Sreya Roy C, et al. Antipoisoning catalysts for the selective oxygen reduction reaction at the interface between metal nanoparticles and the electrolyte[J]. Carbon Energy, 2023, e293.
[54] Kedhareswara Sairam P, Sayandeep G, Nagabandi J, et al. Boron doped g-C3N4 quantum dots based highly sensitive surface acoustic wave NO2 sensor with faster gas kinetics under UV light illumination[J]. Sensors and Actuators B: Chemical, 2022, 378, 133140.
List of changes:
- Line 13, ‘engineer’ reminds to ‘engineering’; Line 92, ‘metals’ was added. And some other mistakes were also corrected and marked.
- In case of misunderstanding, the ‘full-light’ in the article has revised to ‘solar light’.
- Line 82-83, “And on the other hand, they are in favor of improving visible light absorption influenced and promot photoinduced charges separation and active oxygen species production (·O2-) by the surface plasmon resonance effect (SPR)”.
- Line 90-91, Meanwhile, the addition of heat can promote the SMSI effect to promoted the oxygen activation in thermaocatalytic oxidation process, so that the pollutants can be degradated quickly on the surface of the carrier and precious metals.
- Line 166, Line 200 and Line 216, ‘Figure’ reminds to ‘’
- Line 281-282, Line 248, ‘ 4c, d and f’ were revised with error bar.
- Line 171-173, “Due to the structural differences of precursors, there are obvious differences in morphology, interplanar packing space and crystallinity of g-C3N4 synthesized in the thermal polymerization process.” was added in the revised manuscript.
- Line 199 and Line 279, the Fig. 2 and Fig. 5 has revised in the manuscript.
- Reference 13, 31, 52, 54 were added in the revised manuscript.
Round 2
Reviewer 2 Report
Section 4, change "conclusions" to "conclusion"
-Thoroughly read the manuscript and Double-check for any typos
Author Response
We would like to thank you very much for handling our manuscript (ijms-2285523). According to the reviewers’ comments, we have thoroughly read and checked the manuscript. The changes in the manuscript are grey marked for your convenience.
List of changes
1. Line 14, ‘catalysts’ was reminded to ‘catalyst’; Line 75 ‘degration’ was reminded to ‘degradation’; Line 147, ‘temperacture’was reminded to ‘temperature’.
2. Line 154, ‘qurtze’ was revised to ‘quartz’; Line 258, ‘effciency’ was revised to ‘efficiency’.
3. Line 326, ‘conclusions’ was revised to ‘conclusion’.